# Pd/Pt-Bimetallic-Nanoparticle-Doped In_2_O_3_ Hollow Microspheres for Rapid and Sensitive H_2_S Sensing at Low Temperature

**DOI:** 10.3390/nano13040668

**Published:** 2023-02-08

**Authors:** Kaisheng Jiang, Tingting Chen, Jianhai Sun, Hao Quan, Tianye Zhou

**Affiliations:** 1State Key Laboratory of Transducer Technology, Aerospace Information Research Institute, Chinese Academy of Sciences, Beijing 100194, China; 2School of Electronic, Electrical and Communication Engineering, University of Chinese Academy of Sciences, Beijing 100049, China

**Keywords:** hydrogen sulfide gas sensor, In_2_O_3_ hollow microspheres, Pd/Pt bimetallic catalyst

## Abstract

H_2_S is a poisonous gas that is widespread in nature and human activities. Its rapid and sensitive detection is essential to prevent it from damaging health. Herein, we report Pd- and Pt-bimetallic-nanoparticle-doped In_2_O_3_ hollow microspheres that are synthesized using solvothermal and in situ reduction methods for H_2_S detection. The structure of as-synthesized 1 at% Pd/Pt-In_2_O_3_ comprises porous hollow microspheres assembled from In_2_O_3_ nanosheets with Pd and Pt bimetallic nanoparticles loaded on its surface. The response of 1 at% Pd/Pt-In_2_O_3_ to 5 ppm H_2_S is 140 (70 times that of pure In_2_O_3_), and the response time is 3 s at a low temperature of 50 °C. In addition, it can detect trace H_2_S (as low as 50 ppb) and has superior selectivity and an excellent anti-interference ability. These outstanding gas-sensing performances of 1 at% Pd/Pt-In_2_O_3_ are attributed to the chemical sensitization of Pt, the electronic sensitization of Pd, and the synergistic effect between them. This work supplements the research of In_2_O_3_-based H_2_S sensors and proves that Pd- and Pt-bimetallic-doped In_2_O_3_ can be applied in the detection of H_2_S.

## 1. Introduction

Hydrogen sulfide (H_2_S) is a common gas. In addition to its natural source, a large amount of H_2_S is also produced in petroleum refining, sewage treatment, textile treatment, and other human activities [1]. However, H_2_S is highly poisonous and corrosive. Exposure to H_2_S at concentrations above 250 ppm can be fatal. Exposure to low concentrations of H_2_S (10 ppm) also stimulates the visual and olfactory systems, resulting in tears, dizziness, etc. The safety critical concentration of H_2_S in relevant industries is specified as 10 ppm [2]. Therefore, the sensitive and rapid detection of H_2_S is critical to ensure the safety and preservation of relevant workers’ health. According to different mechanisms, gas sensors can be divided into surface acoustic wave sensors [3], chemosensitive-based sensors [4], and metal oxide semiconductor (MOS) sensors, etc. Among them, metal oxide semiconductors, such as SnO_2_, In_2_O_3_, TiO_2_, and IGZO [5,6,7,8,9], are some of the most extensively studied gas-sensing materials, and they have become widely used in gas sensors across the world due to their advantages of a low cost, small size, and high response [10].

Indium oxide (In_2_O_3_) is a typical wide-band-gap (3.5–3.7 eV) n-type semiconductor [11], and it has been utilized in various applications, such as lithium-ion batteries [12] and photocatalysis [13]. For gas sensing, the advantages of a wide band gap and a high conductivity make In_2_O_3_ have potential in the detection of H_2_, HCHO, and CO [14,15,16]. However, the research on In_2_O_3_-based H_2_S sensors is still relatively scarce. Pristine In_2_O_3_-based gas sensors have the disadvantages of a low response, poor selectivity, and a high operating temperature [17]. Numerous studies have proven that synthesizing two-dimensional (2D) or three-dimensional (3D) nanocomposites and doping metals are effective methods to improve the gas-sensing performance of In_2_O_3_.

The 3D structures assembled from 2D nanosheets can maintain the ultra-thin structures of nanosheets while avoiding the performance loss caused by the stacking of the nanosheets [18]. Nanosheet-assembled structures are porous and have large specific surface areas. A larger specific surface area means that there are more atoms on the surface of the material, which can provide more gas adsorption sites and make more gas molecules interact with the material to enhance the gas-sensing property. Liu et al. [19] synthesized an In_2_O_3_ porous pompon assembled from 2D nanosheets using the hydrothermal method, and its response to 50 ppm ethanol was 3.4. Qin et al. [20] synthesized porous In_2_O_3_ nanosheet-assembled microflowers, which had a response of 66 to 100 ppm ethanol and could respond (12.4 s) and recover (10.4 s) quickly. This was because the porous structures provided more gas adsorption sites and allowed gas molecules to diffuse to the depth of the material. Doping metal can change the carrier concentration in In_2_O_3_ by improving the properties of the electron donors or acceptors, changing the structure of In_2_O_3_ or sensitization to enhance the gas-sensing performance [21,22]. Platinum (Pt) and palladium (Pd) are important dopants because Pt possess strong catalytic performance, and Pd can change the carrier concentration of a material through electron sensitization [21]. Chen et al. [23] synthesized Pd-In_2_O_3_ nanofibers via electrospinning, and their response to 10,000 ppm H_2_ at room temperature was 146.8 times that of pure In_2_O_3_; this was because Pd provided more O_2_ absorption sites and could form a Schottky barrier with In_2_O_3_. Song et al. [24] improved the response of a sensor to 10 ppm acetone from 12 to 113 by modifying Pt on the surface of In_2_O_3_ nanotubes, which benefited from more adsorbed oxygen molecules because of the catalytic effect of Pt. On the basis of monometallic catalysts, some studies have shown that, when several metals are doped into a material in a specific proportion, the gas-sensing property of the material is further improved compared to that of a material doped with a single metal. In bimetallic catalysts, the two metals may exhibit synergistic effects due to geometric effects, the interaction of electrons, or chemical reactions, and they exhibit a better gas-sensing property than monometallic catalysts [25,26]. The work of Huang et al. [15] showed that the response of In_2_O_3_ modified by Ag nanoparticles (NPs) and Au nanocages (NCs) to 100 ppm HCHO could reach 3400. The response of Ag NP-In_2_O_3_ was 519, and that of Au NC-In_2_O_3_ was 240 under the same condition. This was mainly because the electrons in Au transferred to Ag, enhancing the spillover effect of Ag.

In this work, we synthesized In_2_O_3_ hollow microspheres doped with Pd and Pt bimetallic nanoparticles (Pd/Pt NPs) using the solvothermal and in situ reduction methods. The structure, morphology, and elemental composition of the microspheres were characterized using scanning electron microscopy (SEM), transmission electron microscopy (TEM), high-resolution transmission electron microscopy (HRTEM), nitrogen adsorption and desorption, X-ray diffraction (XRD), and X-ray photoelectron spectroscopy (XPS). Afterward, a sensor based on this material was prepared, and a gas test was carried out. The working temperature, response/recovery time, dynamic characteristics, limit of detection (LOD), repeatability, and selectivity of the sensor were investigated via the gas tests. The response of the as-synthesized 1 at% Pd/Pt-NP-doped In_2_O_3_ hollow microspheres (1 at% Pd/Pt-In_2_O_3_) to 5 ppm H_2_S at 50 °C was 140, which is 70 times that of the pure In_2_O_3_. The response time of the sensor was 3 s, and the LOD was 50 ppb; moreover, the sensor had a strong linearity, superb repeatability, and great selectivity. However, the recovery time of the sensor was relatively long. It was necessary to increase the working temperature of the sensor to 400 °C through a voltage pulse to speed up the desorption process. The resistance of the sensor could quickly recover to the original value after removing the voltage pulse. In addition, the sensing mechanism of the sensor was also analyzed. The chemical sensitization of Pt, the electronic sensitization of Pd, and the synergistic effect of Pd and Pt were the important factors underlying the excellent gas-sensing performance of the sensor.

## 2. Materials and Methods

### 2.1. Chemicals

All reagent-grade chemicals, including indium nitrate hydrate (In(NO_3_)_3_·4.5H_2_O), platinum chloride (PtCl_4_), palladium chloride (PdCl_2_), isopropanol, glycerol, and ascorbic acid, were purchased from Sinopharm Chemical Reagent Co., Ltd., Beijing, China, and they were used without further purification.

### 2.2. Synthesis of Pd/Pt Bimetallic Nanoparticles Doped In_2_O_3_ Hollow Microspheres

In_2_O_3_ hollow microspheres were synthesized using the solvothermal method. First, 0.6 g indium nitrate hydrate (In(NO_3_)_3_·4.5H_2_O) was added into a beaker containing 60 mL isopropanol and 20 g glycerin. Subsequently, the solution was vigorously stirred at room temperature for 3 h. The obtained solution was transferred into a Teflon-lined stainless-steel autoclave for reaction at 180 °C for 6 h. After naturally cooling to room temperature, the products were collected via centrifugation at a speed of 3500 r/min, and they were washed with absolute ethanol three times. The white In_2_O_3_ precursors were obtained after drying at 60 °C for 24 h. The precursors were then transferred to a ceramic crucible and calcined at 400 °C for 6 h in an air atmosphere in order to obtain the In_2_O_3_ hollow microspheres.

The doping of Pd/Pt NPs was completed via an in situ reduction. A total of 0.2 g of the In_2_O_3_ hollow microspheres synthesized in the first step was added to 40 mL deionized water. The canary color solution was obtained after carrying out ultrasonic irradiation for 15 min and vigorous stirring at room temperature for 3 h. Pt with a content of 1 at% and Pd with a content of 1.8 at% were added to the solution in turn. After 15 min of ultrasonic irradiation, the solution was stirred until there was no obvious sediment in the beaker. Slightly excess ascorbic acid solution (0.01 M) was then slowly dropped into the beaker. The solution was vigorously stirred at room temperature for 3 h after it turned black. The products were collected via centrifugation at a speed of 3500 r/min, and they were washed with deionized water and absolute ethanol three times. Finally, 1 at% Pd/Pt-In_2_O_3_ was obtained after drying at 60 °C for 24 h.

### 2.3. Material Characterization

The changing trend of the mass of the samples with temperature was characterized using a thermal gravimetric analyzer (TG, NETZSCH STA 449 F5/F3 Jupiter, NETZSCH, Selb, Germany). The surface morphology and microstructure of the samples were observed using scanning electron microscopy, transmission electron microscopy, and high-resolution transmission electron microscopy (SEM, TEM, and HRTEM, respectively; SU8020, Hitachi, Tokyo, Japan, and FEI Tecnai G2 F30, FEI, Hillsboro, OR, USA). The specific surface area and pore size of the as-synthesized samples were determined via nitrogen adsorption–desorption isotherms (Micromeritics TriStar III 3020, Micromeritics Instrument Corporation, Norcross, GA, USA). The composition of the material and the phase composition of the samples were determined using X-ray diffraction (XRD, Bruker D8 Advance, Bruker, Billerica, MA, USA) with CuKα Irradiation (λ = 0.154 nm). The scanning range (2θ) was 10–80°, and the scanning speed was 6 °/min. The elemental and combined state composition of the surface of the samples were analyzed using X-ray photoelectron spectroscopy (XPS, Thermo Scientific Escalab 250Xi, Thermo Fisher Scientific, Waltham, MA, USA). All binding energies were calibrated according to the C 1s peak of surface indeterminate carbon at 284.8 eV.

### 2.4. Gas Sensor Fabrication and Measurement

A gas sensor was obtained by coating the gas-sensing material on a micro-hotplate. The micro-hotplate is a kind of microheater prepared using MEMS technology. It consists of a substrate layer, a heating layer, an insulating layer, and a testing layer. The heating layer comprises a heating circuit wire and a pair of heating electrodes composed of Pt. The insulating layer comprises an insulating medium that separates the heating layer from the test layer. The test layer comprises an interdigital electrode and a pair of test electrodes composed of Au wires. When the sensor is working, the heating layer provides the corresponding temperature for the gas-sensing material coated on the test layer according to the voltage applied to the heating electrodes. The interdigital electrode on the test layer will be conductive after being coated with the gas-sensing material, and the resistance change in the gas-sensing material can be detected through the test electrodes.

The gas sensor was prepared in the following steps: The as-synthesized 1 at% Pd/Pt-In_2_O_3_ was mixed with an appropriate amount of anhydrous ethanol. The uniformly mixed paste was uniformly coated on the interdigital electrode in the center of the test layer. After the paste dried, the micro-hotplate and the base were combined to obtain a gas sensor. The static gas distribution method was used to measure the gas response of the sensor. The sensor was aged at 100 °C for 24 h to improve its stability before gas tests. A static gas distribution method was applied to obtain the desired concentration of the target gas in the test chamber, and the air was used as the background gas. The response of the sensor was calculated by detecting the resistance change in the sensor. The response of the sensor (*S*) is defined according to Equation (1):*S = R_a_*/*R_g_*,
(1)
where *R_a_* and *R_g_* are the resistances in the air and the target gas, respectively. The response time (T_Response_) and recovery time (T_Recovery_) of the sensor are defined as the time spent when the sensor resistance changes from the initial value to 90% of the total resistance change after the target gas is applied and released.

## 3. Results

### 3.1. Results of Characterization

A thermogravimetric (TG) analyzer was used to detect the change in the quality of the precursors with the increase in the calcination temperature. According to Figure 1, the quality of the precursors drops sharply in the range of 230–360 °C, which is mainly due to the thermal decomposition of the organic components in the precursors. Therefore, the calcination temperature was set to 400 °C.

The surface morphology and microstructure of 1 at% Pd/Pt-In_2_O_3_ were observed using FESEM, TEM, and HRTEM, as shown in Figure 2. It can be observed from the low-resolution SEM image in Figure 2a that the material is composed of many ellipsoidal microspheres with a rough surface, and the size of the microspheres is about 600–900 nm. It can be further observed from the high-resolution SEM image shown in Figure 2b that the microspheres are porous structures assembled from many nanosheets, with Pd/Pt NPs loaded on the surfaces of the nanosheets. It can be inferred from the broken sphere in the image that the microspheres are hollow. The TEM image shown in Figure 2c further confirms the above results: the microspheres are hollow structures, and flocculent structures extend from the surface of the spherical shell. The microsphere structures, especially the flocculent structures, are porous and loose. Figure 2d presents the HRTEM image of 1 at% Pd/Pt-In_2_O_3_. Two groups of lattice fringes with distances of 0.25 nm and 0.29 nm can be observed, which are attributed to the (400) and (222) crystal planes of cubic In_2_O_3_, respectively [27].

The specific surface area and pore size of the pure In_2_O_3_ and 1 at% Pd/Pt-In_2_O_3_ were characterized via nitrogen adsorption and desorption isotherms. In the nitrogen adsorption and desorption isotherms presented in Figure 3a, a hysteresis loop is observed in the isotherms of both materials, indicating mesoporous structures in the microspheres [25]. The specific surface area of the material was measured using the Brunauer–Emmett–Teller method (BET). The specific surface area of the pure In_2_O_3_ is 61.5061 m^2^/g, while that of 1 at% Pd/Pt-In_2_O_3_ is 58.3404 m^2^/g. Figure 3b indicates the pore size distribution of the pure In_2_O_3_ and 1 at% Pd/Pt-In_2_O_3_. The pore size of the pure In_2_O_3_ is concentrated in the range of 35–51 nm, while that of 1 at% Pd/Pt-In_2_O_3_ is concentrated in the range of 25–35 nm. The specific surface area and pore size of the doped microspheres decrease slightly because the Pd/Pt NPs loaded on the surface of the microspheres block some of the pores. In general, doping does not significantly affect the gas adsorption capacity of the microspheres.

Figure 3c presents the XRD patterns that were obtained for the pure In_2_O_3_ and 1 at% Pd/Pt-In_2_O_3_ to analyze the crystal structures. The pure In_2_O_3_ has strong diffraction peaks at 21.47°, 30.54°, 35.44°, 50.97°, and 60.65°, which correspond to (211), (222), (400), (440), and (622) of cubic In_2_O_3_ (JCPDS No. 71–2195), respectively. This result proves that the sample contains In_2_O_3_. However, the XRD images of 1 at% Pd/Pt-In_2_O_3_ and the pure In_2_O_3_ are not significantly different. In addition, the diffraction peaks corresponding to Pd and Pt are not observed, probably because of the low content of Pd and Pt beyond the detection limit of the diffractometer.

The elemental and combined state compositions of the surface of the sample were analyzed using XPS, and the results are shown in Figure 4. Figure 4a presents an XPS survey spectrum containing multiple spectral lines of O, In, and C elements. The diffraction peaks corresponding to Pd and Pt are not obvious due to the low content of Pd and Pt. Figure 4b shows the XPS spectra of In 3d. The In 3d peak can be divided into two peaks at 444.2 and 451.8 eV, corresponding to the binding energies of In 3d_3/2_ and In 3d_5/2_ valence states, respectively [28,29]. This result shows that the indium element in the sample is mainly in the form of In^3+^ [30]. Figure 4c illustrates the peak related to the O element. The peak can be divided into 529.6 and 531.2 eV peaks, corresponding to lattice oxygen (O_lattice_) and chemisorbed oxygen O_x_^−^ (O^−^, O_2_^−^, and O^2−^), respectively [31]. After performing calculations according to the peak area, it is found that the proportion of chemisorbed oxygen is about 51.5%. The Pt 4f spectra shown in Figure 4d can be obtained from the high-resolution XPS spectra. It can be divided into two peaks at 70.4 and 76.3 eV, corresponding to the binding energies of Pt 4f_7/2_ and Pt 4f_5/2_ valence states, respectively, assigned to Pt. Figure 4e presents the high-resolution spectra of Pd 3d, and they show two characteristic peaks with binding energies of about 334.8 and 340.4 eV, corresponding to the binding energies of Pd 3d_5/2_ and Pd 3d_3/2_, respectively. This result proves the existence of Pd [32].

### 3.2. Gas-Sensing Characteristics

Figure 5 illustrates the responses of the pure In_2_O_3_ and 1 at% Pd/Pt-In_2_O_3_ to 5 ppm H_2_S at different operating temperatures. Since the minimum working temperature of the micro-hotplate in the sensor is 50 °C, the temperature gradient starts from 50 °C to ensure the stability of the working temperature of the sensor. As shown in Figure 5, when the operating temperature is 50 °C, 1 at% Pd/Pt-In_2_O_3_ has the highest response to H_2_S, and the response decreases with the increase in the operating temperature. The response of the pure In_2_O_3_ to H_2_S increases with the temperature, and it reaches the maximum at 250 °C. The optimal operating temperature of 1 at% Pd/Pt-In_2_O_3_ is significantly lower than that of the pure In_2_O_3_ because of the catalysis of Pt [33]. In addition, the response of 1 at% Pd/Pt-In_2_O_3_ (140) is close to 70 times that of the pure In_2_O_3_ (2.1) at 50 °C, which indicates that the introduction of Pd/Pt NPs significantly improves the gas sensitivity of the microspheres to H2S. The electronic sensitization of Pd, the chemical sensitization of Pt, and the synergistic effect between them are the main reasons for the enhanced response of 1 at% Pd/Pt-In_2_O_3_.

Figure 6 shows the response/recovery time of 1 at% Pd/Pt-In_2_O_3_ for 5 ppm H_2_S at 50 °C. The response time (T_Response_) and recovery time (T_Recovery_) of the sensor are defined as the time spent when the sensor resistance changes from the initial value to 90% of the total resistance change after the target gas is applied and released. The sensor responds very quickly after contacting H_2_S, and the response time is only 3 s (T_Response_ = 3 s). According to the characterization, the structure of 1 at% Pd/Pt-In_2_O_3_ is porous and has a specific surface area of 58.3704 m^2^/g, which enables the gas to diffuse into the interior of the material; there are many active sites on the surface of the material where hydrogen sulfide can be bound, which accelerates the response speed [20]. In addition, a large amount of chemically adsorbed oxygen is distributed on the material surface due to the overflow effect of the Pt NPs, which further shortens the response time [25]. However, after H_2_S is released, the sensor recovers very slowly. After 1000 s, the resistance of the sensor only recovers from 10% to 28% of the original resistance. This slow recovery may be related to the existence of many cracks and pores in the NP-loaded materials, which hinder the desorption of the H_2_O and SO_2_ generated in the reaction between the H_2_S and the chemically adsorbed oxygen [34,35]. To solve this problem, a voltage pulse is applied to the sensor to increase the working temperature of the sensor to 400 °C in order to promote the desorption of H_2_S. As shown in Figure 6, the sensor resistance value surges after the pulse is applied, and then it quickly returns to the original value. This proves that the application of a voltage pulse can effectively shorten the recovery time of the sensor because the high temperature accelerates the desorption of H_2_O and SO_2_.

Figure 7a shows the dynamic change in the resistance and response of 1 at% Pd/Pt-In_2_O_3_ to different concentrations of H_2_S (500 ppb–10 ppm) at 50 °C. The baseline resistance of the sensor is not affected by the concentration of H_2_S, and the response of the sensor increases significantly with the increase in the concentration of H_2_S. Figure 7b shows the relationship between the response of 1 at% Pd/Pt-In_2_O_3_ and the concentration of H_2_S. The response of the sensor has a strong linear relationship with the logarithm of the concentration of H_2_S, which indicates that the sensor can be applied to H_2_S sensing. As shown in Figure 7c, the response of the sensor to 50 ppb H_2_S is 1.6 at 50 °C. Therefore, the LOD of the sensor is 50 ppb.

Figure 8a shows the effect of humidity on the H_2_S sensing of 1 at% Pd/Pt-In_2_O_3_. The response value of the sensor decreases significantly with the increase in humidity. When the relative humidity increases from 25% to 75%, the response of the sensor decreases by 47%. The change trend of the response of 1 at% Pd/Pt-In_2_O_3_ with humidity and that of the normalized resistance of a humidity sensor based on metal oxide are similar, so the performance loss may be caused by the influence of the humidity on the resistance of the sensor [36]. In addition, the occupation of some active sites by H_2_O molecules affects the adsorption of gas on the surface of the material and causes a further decline in the response [37].

Figure 8b indicates the response of the sensor to five groups of 5 ppm H_2_S at 50 °C. The responses of the sensor to the five groups of introduced H_2_S are the same, and their standard deviation is 1.4519, demonstrating the superb repeatability of the sensor.

Figure 8c presents the response of 1 at% Pd/Pt-In_2_O_3_ to various gases. When the working temperature is 50 °C, the response of the sensor to 5 ppm H_2_S is significantly higher than its response to various gases with concentrations of 50 ppm. Therefore, the sensor has a strong selectivity and can effectively avoid the influence of interference gas.

The long-term stability of 1 at% Pd/Pt-In_2_O_3_ is measured by repeatedly testing its response to 5 ppm H_2_S at 50 °C within 20 days. According to Figure 8d, the response of the sensor is almost stable within 20 days and has potential for practical applications.

In conclusion, 1 at% Pd/Pt-In_2_O_3_ is characterized by a low operating temperature, a high sensitivity (140 for 5 ppm H_2_S), a fast response, a low LOD, a linear response, superior selectivity, and repeatability. Table 1 demonstrates a comparison of the H_2_S sensing performance of In_2_O_3_-based sensors in the recently reported literature and in our work. The response time of 1 at% Pd/Pt-In_2_O_3_ to 5 ppm H_2_S at 50 °C is 3 s, which is much faster than that of the other materials listed in the table. Additionally, the response of 1 at% Pd/Pt-In_2_O_3_ under the same conditions is 140. The other materials have either a higher operating temperature or a lower response at 50 °C. Therefore, the gas-sensing material synthesized in this paper has an excellent low-temperature performance and can be used for the rapid and sensitive detection of H_2_S at a low temperature.

### 3.3. Gas-Sensing Mechanism

The gas-sensing mechanism of In_2_O_3_ is a surface-controlled type, as shown in Figure 9 [43]. When In_2_O_3_ belonging to an n-type semiconductor is exposed to air, the oxygen in the air will adsorb on the surface of In_2_O_3_ to form adsorbed oxygen, as shown in Equation (2). The adsorbed oxygen will capture the conduction-band electrons on the surface of In_2_O_3_ to generate chemisorbed oxygen, as shown in Equations (3)–(5) [44]. The types of chemisorbed oxygen that form are different according to the different ambient temperatures [45]. As electrons are trapped, the thickness of the electronic depletion layer increases, the carrier mobility of the surface of In_2_O_3_ decreases, and the resistance increases.
O_2_ (gas) → O_2_ (ads),(2)
O_2_ (ads) + e^−^ → O_2_^−^ (ads) T < 147 °C,(3)
O_2_^−^ (ads) + e^−^ → 2O^−^ (ads) 147 °C < T < 397 °C,(4)
O^−^ (ads) + e^−^ → O^2^^−^ (ads) T > 397 °C,(5)

Since the working temperature of 1 at% Pd/Pt-In_2_O_3_ is 50 °C, the type of chemisorbed oxygen should be mainly O_2_^−^ [46]. When the reductive H_2_S contacts the chemisorbed oxygen, H_2_S will react with the chemisorbed oxygen, as shown in Equation (6).
2H_2_S (g) + 3O_2_^−^ → 2H_2_O (g) + 2SO_2_ + 3e^−^,(6)

In this reaction, the electrons captured by the chemisorbed oxygen return to the conduction band of In_2_O_3_. The thickness of the electronic depletion layer decreases, the carrier mobility of In_2_O_3_ recovers, and the resistance decreases.

According to the measurements of the specific surface areas of the microspheres, the specific surface area of 1 at% Pd/Pt-In_2_O_3_ is slightly smaller than that of the pure In_2_O_3_. In general, the gas-sensing performances of materials with larger specific surface areas are enhanced. However, the response of 1 at% Pd/Pt-In_2_O_3_ to H_2_S is much higher than that of the pure In_2_O_3_, which is contrary to the regular pattern. We believe that this is because the introduction of Pd/Pt NPs can improve the gas-sensing performance of the material much more than the loss caused by a reduction in the specific surface area, which demonstrates that the introduction of Pd/Pt NPs can effectively improve the gas-sensing properties of materials. Pt NPs mainly improve gas sensitivity through chemical sensitization. Pt can promote the formation of chemisorbed oxygen through chemical sensitization. Due to the influence of Pt, a large amount of chemisorbed oxygen overflows to the surface of the material, which improves the sensitivity of the material [33]. Pd improves the gas-sensing property through electronic sensitization. As the work function of Pd NPs (5.5 eV) is greater than that of In_2_O_3_ (5.0 eV) [47], a Schottky barrier forms between the Pd NPs and In_2_O_3_. Compared to monometallic catalysts, Pd/Pt bimetallic NPs can further improve the catalytic activity because of the synergistic effects between them. As shown in Equation (6), water is generated in the reaction of H_2_S and chemically adsorbed oxygen. It reacts with Pd to form Pd(OH)_2_ with low catalytic activity, which reduces the activity of Pd [48]. Doping an appropriate amount of Pt into a material can inhibit this process and reduce the influence of water on Pd [49].

## 4. Conclusions

In summary, In_2_O_3_ hollow microspheres doped with Pd/Pt NPs were successfully synthesized using the solvothermal method and the in situ reduction method. Due to the excellent physical properties of In_2_O_3_ and the sensitization and synergistic effect of the dopants, the gas sensor based on 1 at% Pd/Pt-In_2_O_3_ has the advantages of a low operating temperature, a high sensitivity, a low LOD, a fast response, superior selectivity, etc. Specifically, the response of the sensor to 5 ppm H_2_S at 50 °C can reach 140, and the response time is only 3 s. The disadvantage of the slow recovery is also solved by applying a voltage pulse during the recovery process. The material proposed in this paper has potential application prospects in H_2_S sensors.

## Figures and Tables

**Figure 1 nanomaterials-13-00668-f001:**
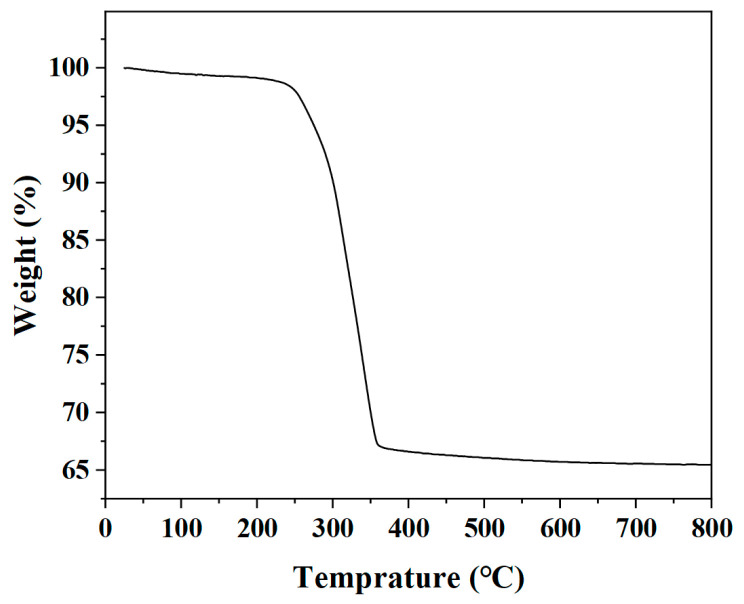
TG curve of the precursors of In_2_O_3_ hollow microspheres.

**Figure 2 nanomaterials-13-00668-f002:**
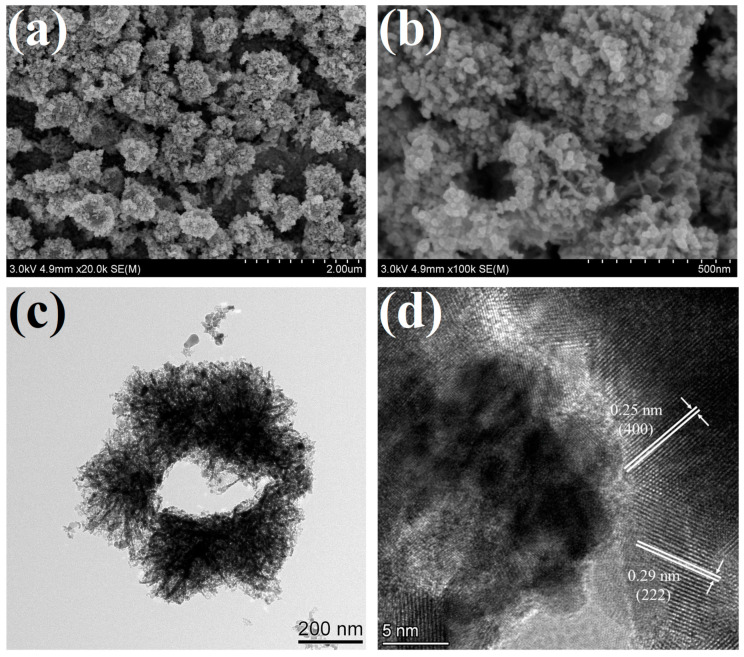
SEM and TEM images of 1 at% Pd/Pt-In_2_O_3_: (**a**) low-resolution SEM image, (**b**) high-resolution SEM image, (**c**) TEM image, (**d**) HRTEM image.

**Figure 3 nanomaterials-13-00668-f003:**
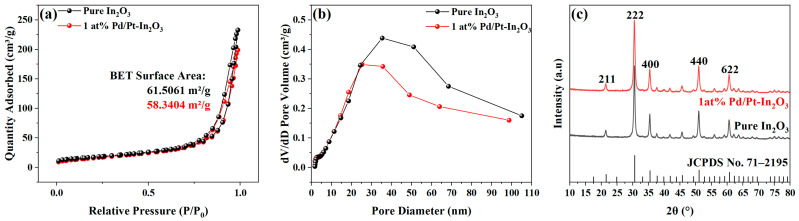
(**a**) Nitrogen adsorption−desorption isotherms of pure In_2_O_3_ and 1 at% Pd/Pt-In_2_O_3_. (**b**) Corresponding pore size distributions of pure In_2_O_3_ and 1 at% Pd/Pt-In_2_O_3_. (**c**) XRD image of pure In_2_O_3_ and 1 at% Pd/Pt-In_2_O_3_.

**Figure 4 nanomaterials-13-00668-f004:**
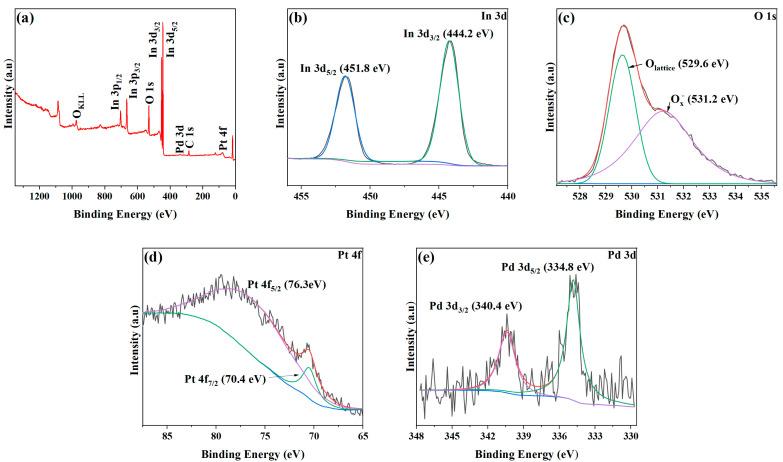
XPS spectra of 1 at% Pd/Pt-In_2_O_3_. (**a**) Survey spectrum and high-resolution XPS spectra: (**b**) In 3d, (**c**) In 3d, (**d**) Pt 4f, and (**e**) Pd 3d.

**Figure 5 nanomaterials-13-00668-f005:**
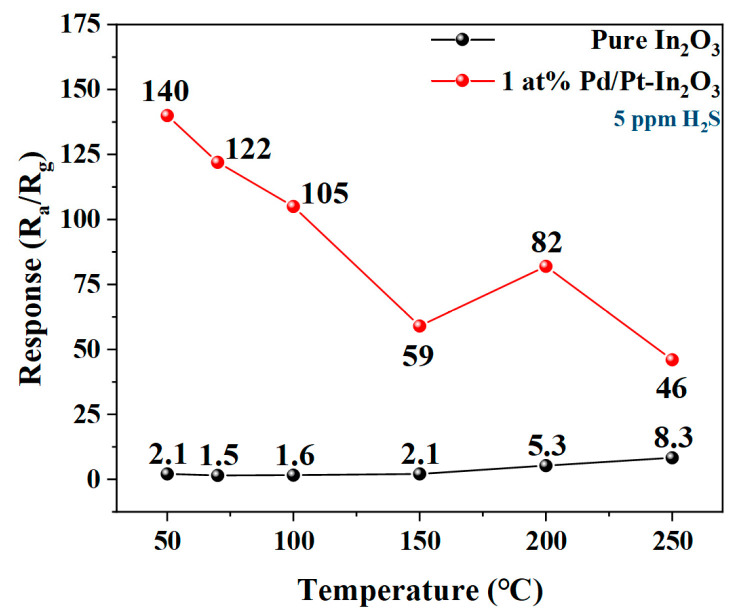
Response of pure In_2_O_3_ and 1 at% Pd/Pt-In_2_O_3_ to 5 ppm H_2_S at different temperatures.

**Figure 6 nanomaterials-13-00668-f006:**
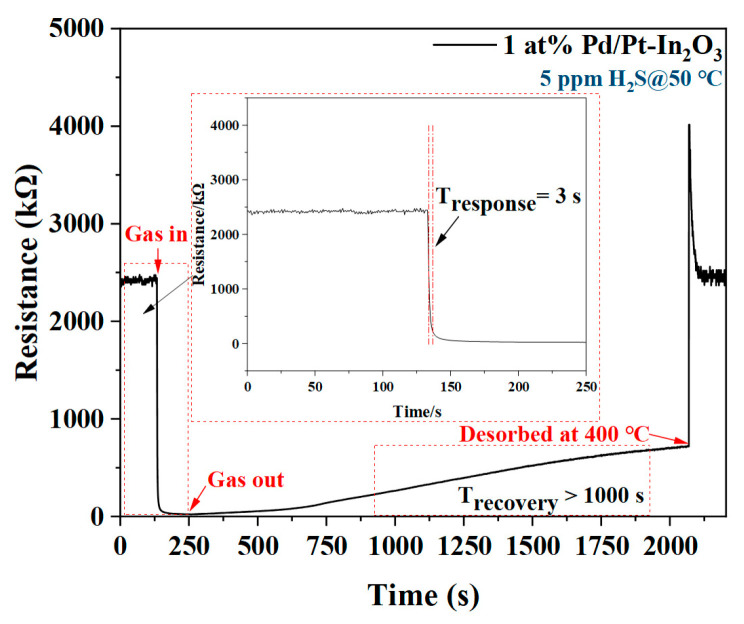
Response/recovery time of 1 at% Pd/Pt-In_2_O_3_ for 5 ppm H_2_S at 50 °C.

**Figure 7 nanomaterials-13-00668-f007:**
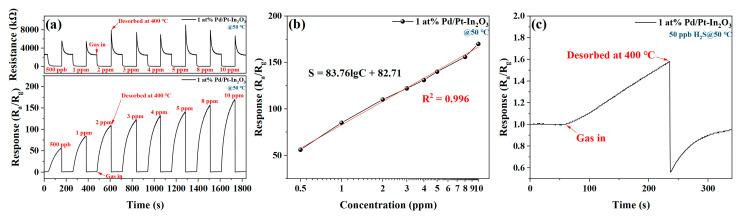
(**a**) Dynamic change in resistance and response of 1 at% Pd/Pt-In_2_O_3_ to different concentrations of H_2_S (500 ppb–10 ppm) at 50 °C. (**b**) Relationship between the response of 1 at% Pd/Pt-In_2_O_3_ and the concentration of H_2_S at 50 °C. (**c**) Response of 1 at% Pd/Pt-In_2_O_3_ to 50 ppb H_2_S at 50 °C.

**Figure 8 nanomaterials-13-00668-f008:**
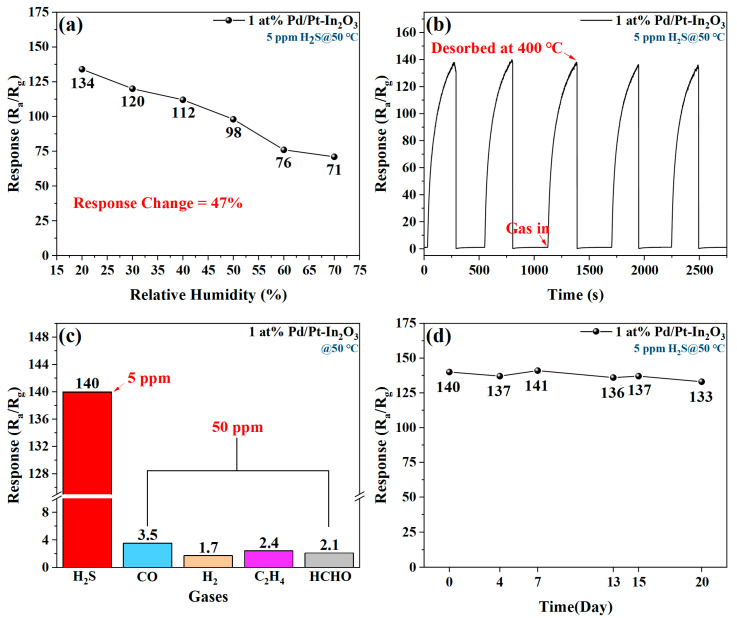
(**a**) Effect of humidity on H_2_S sensing of 1 at% Pd/Pt-In_2_O_3_. (**b**) Five-time repeatability of 1 at% Pd/Pt-In_2_O_3_ to 5 ppm H_2_S at 50 °C. (**c**) Response of 1 at% Pd/Pt-In_2_O_3_ to various gases (CO, H_2_, C_2_H_4_, and HCHO). (**d**) Long-term stability of 1 at% Pd/Pt-In_2_O_3_.

**Figure 9 nanomaterials-13-00668-f009:**
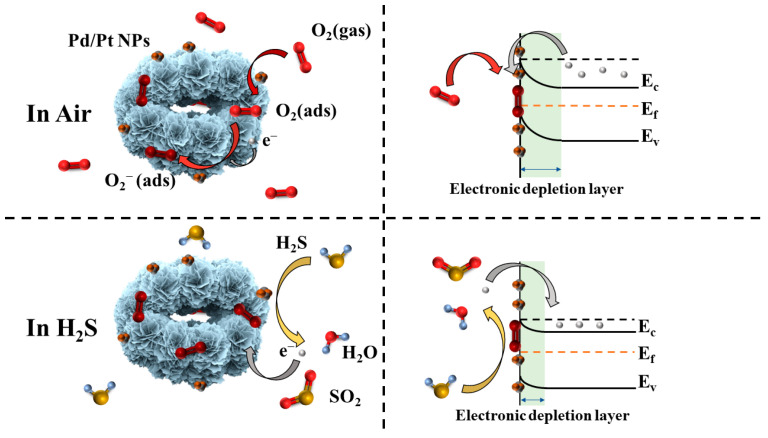
Demonstration of gas-sensing mechanism of 1 at% Pd/Pt-In_2_O_3_. The left side is a simulation diagram of the microsphere, and the right side is an energy-band diagram.

**Table 1 nanomaterials-13-00668-t001:** The comparison in H_2_S sensing performance of In_2_O_3_-based sensors between the reported literatures and our work.

Materials	Temp. (°C)	Conc. (ppm)	Res.	Res./Rec. Time (s/s)	LOD (ppm)	Refs.
Pd/Pt-In_2_O_3_ hollow micromicrospheres	50	5	140	3/-	50 ppb	This work
Cu-In_2_O_3_ hollow nanofibers	250	100	4201.5	30/18	1	[34]
Bamboo-like CuO/In_2_O_3_ heterostructures	70	5	229.3	10/-	0.2	[37]
Li_0.5_La_0.5_TiO_3_-In_2_O_3_ nanorods	150	50	116.61	20/287	5	[38]
Ce_2_O_3_/In_2_O_3_ hollow microspheres	100	20	3.8	59/600	0.1	[39]
Ag/In_2_O_3_ porous hexagonal tubes	70	5	119	20/-	0.3	[40]
In_2_O_3_/ZnO porous hollow nanocages	200	50	67.5	52/198	2	[41]
In_2_O_3_ nanoparticles	25	0.1	18.1	60/-	-	[42]

## Data Availability

Not applicable.

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
