# Peer review of "Pd/Pt-Bimetallic-Nanoparticle-Doped In_2_O_3_ Hollow Microspheres for Rapid and Sensitive H_2_S Sensing at Low Temperature"

_nanomaterials, 2023, doi:10.3390/nano13040668_

Round 1

Reviewer 1 Report

This is an interesting and comprehensively-studied research. I would recommend this paper be published as it is due to its high quality.

Author Response

Dear Reviewer,

Thank you very much for your recognition and appreciation of our work. On the occasion of the Chinese New Year, we wish you a happy New Year.

Thanks,

Jianhai Sun

Reviewer 2 Report

In this article, Kaisheng Jiang et al., investigated the H2S gas sensing at low temperatures using sensitive Pd/Pt bimetallic nanoparticles doped In2O3 hollow microspheres. The as-prepared sensor showed high sensitivity, short response/recovery times, good selectivity, and anti-interference ability. The stated results are interesting and the manuscript is also well organized/written, which will likely to attract attention and impact in low-temperature sensing applications. Also, the authors have explained the possible gas-sensing mechanism using band analysis which is interesting. As such, I believe that the manuscript is suitable for publication; in which following minor amendments needed to be clarified before this work can be finally accepted in Nanomaterials journal:

  1. What is the thickness of the sensing layer?
  2. Please give the carrier concentration and conductivity of In2O3 before and after doping in the revised manuscript.
  3. In the HRTEM image (Fig. 2d), the darker particle might be ascribed to the Pt/Pd bimetallic particles. Please make sure and circle the black spots for better understanding to readers.
  4. What are the detection limit and noise levels of the as-prepared composite sensor? Please provide it in the revised manuscript.
  5. It is acceptable that the pore size is decreased due to the addition of Pt/Pd which can block some pores. However, please explain the reason for decreasing the surface area of the composite sample and how it is good for high sensing response in the revised manuscript. In some reports, it was stated as the higher the surface area, the higher the response.
  6. Please explain the reasons for spikes in sensing responses.
  7. It could be better if the authors could compare the In 3d XPS peak before and after doping of In2O3 with Pt/Pd. The core-level binding energy differences in dopant samples can lead to the charge transferring between the nanocomposites.  
  8. How about long-term stability?
  9. Why the resistance and response curves of the same sensor to the same gas is different to look.
  10. The authors have focused only on (CO, H2, 289 C2H4 and HCHO). Have the authors checked some oxidizing agents like NOx?

11.    Some of the references used in the manuscript is not up-to-date; authors should cite the following reference: 10.1016/j.jhazmat.2021.128174, 10.1016/j.snb.2022.131786, 10.1039/D2NJ04117K, 10.1021/acsomega.2c03897.

Author Response

Dear Reviewer,

We gratefully appreciate the your kind suggestions to improve our work. We have studied the comments carefully and tried our best to improve the manuscript. Changes have been made in the revision according to the your comments. We sincerely hope our revision could meet your queries and are welcome further queries. Please feel free to contact us with any further questions.

Thanks,

Jianhai Sun

Reviewer 3 Report

The authors report resistive type gas sensor titled “Pd/Pt Bimetallic Nanoparticles Doped In2O3 Hollow Micro-spheres for Rapid and Sensitive H2S Sensing at Low-tempera- 3 ture”. The reviewer believes that your findings are of great value in this area and will be accepted for publication with a minor revision to reflect some comments.

Comment #1: The description of most of the experiments is too monotonous, and there is no physical and chemical analysis of the data. Make up for this part.

Comment #2: According to your mention in the introductory part, there is a issue at higher concentration of 250 ppm as well, so the reviewer suggest you add a sensitivity characteristic helping the reader to understand your devices.

Comment #3: I agree with you about having a 50oC lower operation temperature, but room temperature operation is still a very important issue in the field. Any additional ideas or directions for improvement to achieve this?

Minor comments;

Comment #1: Please add some more references about recent semiconductor-based works to introduce in the introduction as below.

a.     ACS Sens. 2021, 6, 11, 4217–4224

b.     Journal of the Korean Physical Society volume 80, pages1065–1070 (2022)

c.     ACS Sens. 2022, 7, 9, 2567–2576

d.     Sensors & Actuators: B. Chemical 376 (2023) 132993

Author Response

(The authors gave the same response as above.)

Reviewer 4 Report

1. Since the title highlights the Rapid and Sensitive characteristics of the presented H2S sensors, it is indicated to clarify to a greater extent the response and recovery times as well as the sensitivity compared to those in the literature etc.

2. The effects of humidity on the material and implicitly reflected on the sensors must be clarified because, as can be seen from figure 5, at least at temperatures lower than 150 degrees Celsius, the response for the “red sample” seems to be rather specific to humidity and not to a gas like H2S. Some recent studies related to the influence of humidity on materials and sensors can help in this regard and also can complete the bibliographic references:

https://doi.org/10.1016/j.sna.2021.113060

3. H2S is very corrosive, contributing significantly to the degradation process and aging of the sensor material, which is reflected in a reduced lifetime etc. These effects could have been observed in the measurements of repeatability and long-term stability characteristics, the measurements that are not presented. These aspects should be highlighted in the manuscript. Some recent studies on these aspects can be helpful:

https://doi.org/10.1007/s10854-020-04228-3

4. Regarding the quality of the presentation, the figures must have a unitary aspect, the text must (re)written more carefully etc.

Author Response

(The authors gave the same response as above.)

Round 2

Reviewer 4 Report

...